# Treadmill Running Changes Endothelial Lipase Expression: Insights from Gene and Protein Analysis in Various Striated Muscle Tissues and Serum

**DOI:** 10.3390/biom11060906

**Published:** 2021-06-17

**Authors:** Agnieszka Mikłosz, Bartłomiej Łukaszuk, Adrian Chabowski, Jan Górski

**Affiliations:** 1Department of Physiology, Medical University of Bialystok, 12-222 Bialystok, Poland; bartlomiej.lukaszuk@umb.edu.pl (B.Ł.); adrian.chabowski@umb.edu.pl (A.C.); 2Department of Basic Sciences, Lomza State University of Applied Sciences, 18-400 Lomza, Poland; jan.gorski@umb.edu.pl

**Keywords:** endothelial lipase, HDL-cholesterol, skeletal muscles, heart, diaphragm, treadmill running, rat

## Abstract

Endothelial lipase (EL) is an enzyme capable of HDL phospholipids hydrolysis. Its action leads to a reduction in the serum high-density lipoprotein concentration, and thus, it exerts a pro-atherogenic effect. This study examines the impact of a single bout exercise on the gene and protein expression of the EL in skeletal muscles composed of different fiber types (the soleus—mainly type I, the red gastrocnemius—mostly IIA, and the white gastrocnemius—predominantly IIX fibers), as well as the diaphragm, and the heart. Wistar rats were subjected to a treadmill run: (1) t = 30 [min], V = 18 [m/min]; (2) t = 30 [min], V = 28 [m/min]; (3) t = 120 [min], V = 18 [m/min] (designated: M30, F30, and M120, respectively). We established EL expression in the total muscle homogenates in sedentary animals. Resting values could be ordered with the decreasing EL protein expression as follows: endothelium of left ventricle > diaphragm > red gastrocnemius > right ventricle > soleus > white gastrocnemius. Furthermore, we observed that even a single bout of exercise was capable of inducing changes in the mRNA and protein level of EL, with a clearer pattern observed for the former. After 30 min of running at either exercise intensity, the expression of EL transcript in all the cardiovascular components of muscles tested, except the soleus, was reduced in comparison to the respective sedentary control. The protein content of EL varied with the intensity and/or duration of the run in the studied whole tissue homogenates. The observed differences between EL expression in vascular beds of muscles may indicate the muscle-specific role of the lipase.

## 1. Introduction

Serum lipoproteins are important carriers of lipids in the blood. They provide lipids for cellular oxidative and structural requirements. There are several known types of lipases all of them play an important role in the regulation of lipids levels in the blood. The best known of them are lipoprotein lipase (LPL), hepatic lipase (HL), and endothelial lipase (EL) [1,2]. LPL hydrolyzes serum triglycerides (TG) in the fraction of VLDL and chylomicrons, while HL hydrolyses both TG and phospholipids in the fraction of LDL and HDL, thus it generates smaller LDL and smaller HDL particles. Studies have focused on endothelial lipase (gene nomenclature, LIPG; protein, EL), i.e., a phospholipase with high affinity for phospholipids carried by high-density lipoprotein, and relatively low activity for its (HDL) TGs. As a result, it primarily hydrolyses the first (sn-1) ester bond of phospholipids and releases free fatty acids. It reduces the size of phospholipids in each class of lipoproteins with HDL being a preferred substrate [3,4]. In contrast to other triacylglycerol lipases, EL is synthetized by vascular endothelial cells (thus it has been termed endothelial lipase) and to a lesser extent by smooth muscle cells and macrophages but not by skeletal muscle cells per se [3,5,6]. In general, the human EL expression has been reported to be complementary to what is documented for LPL. Lipoprotein lipase is mainly synthetized in adipocytes, myocytes, and macrophages, whereas HL is synthetized in the liver and in macrophages. Both lipases are then transported to the luminal surface of the endothelial cells, where they hydrolyze TGs and/or phospholipids of lipoproteins. The presence of EL was reported in a variety of different tissues including the placenta, thyroid gland, small intestine, ovary, testis, mammary gland, brain, lung, and aorta, thus suggesting its important role in lipid metabolism [5,7,8,9]. It is also known that imbalances in lipoprotein metabolism contribute to metabolic diseases ranging from vascular inflammation and atherosclerosis to obesity and diabetes. It was found that type 2 diabetes does not affect the expression of EL in human skeletal muscle [10] despite the elevated level of this enzyme in the serum [11]. The EL serum level is significantly increased in humans with metabolic syndrome and associated with atherosclerosis [12]. Its presence was also detected in the foam cells in human atherosclerotic plaques [6,9]. There is evidence indicating that EL exerts pro-atherosclerotic action [3,13,14]. Additionally, the pro-inflammatory cytokines (TNFα, interleukin 1β, and others) and intra-vessel biophysical forces increase the expression of EL in the endothelial cells. Similarly, serum EL concentration in humans is positively correlated with inflammatory markers [7,15,16,17,18,19,20]. It was shown that the before-mentioned TNFα and IL-1β can increase endothelial lipase expression in endothelial cells at both mRNA and protein levels [21,22]. Additionally, it was demonstrated that inhibition of the NfκB-pathway (inflammatory pathway with NfkB as its central hub) abrogated this induction of EL production. These findings suggest that endothelial lipase plays an important role in the pathogenesis of vascular diseases. Furthermore, EL seems to be a major regulator of HDL-cholesterol (HDL-C) serum level. In line with that notion, Robert et al. showed that IL-6 (a well-known pro-inflammatory cytokine) stimulation is capable of increasing EL production by endothelial cells [16]. The above translated into greater HDL translocation through the endothelial wall and a smaller concentration of the lipoprotein in the blood [16]. This seems to be of vital importance since low HDL-C level, i.e., <40 [mg/dL] in women and <50 [mg/dL] in men, is a clinically recognized risk factor for heart disease [23]. Most of the studies show an inverse relationship between the serum HDL-C concentration and its tissue expression [1,3,9,13,14,21,24]. However, in the endurance-trained middle-aged men, but not in the untrained individuals, a positive correlation was found between the expression of the EL protein in the endothelium of vastus lateralis and the concentration of HDL-C in the serum [25]. In addition to phospholipase activity, EL also exerts a non-enzymatic effect, it can promote HDL binding and uptake, as well as the selective uptake of HDL cholesterol esters [4].

Regular exercise is one of the strongest predictors of an individual’s future health and plays an important role in preventing atherosclerosis. Not much is known about the expression and physiological role of EL in the endothelium of muscle vessels. We found only one study investigating the effect of moderate-intensity endurance training (e.g., running or cycling) on the EL protein expression in skeletal muscle tissue. Vigelso et al. demonstrated that trained middle-aged men with high VO_2_max have greater EL protein expression in the endothelium of vastus lateralis in comparison with the untrained individuals. The authors speculated that higher capillarization in the trained muscle is responsible for the enhanced EL expression [25]. However, there are no data regarding the EL expression in the vascular beds of different muscles at rest or after acute exercise. Therefore, we examined the impact of a single exercise bout on the EL gene and protein expression in skeletal muscles as well as in life-long working muscles, i.e., the diaphragm and the heart (the left and right ventricle) homogenates. Different skeletal muscles were chosen for the investigation, since a given skeletal muscle may be comprised of different fiber types (type I, IIA, and IIX), and thus respond distinctively to various metabolic demands. Given the above, we evaluated the EL expression in the predominantly slow-twitch oxidative (type I—soleus), fast-twitch oxidative–glycolytic (type IIA, red section of the gastrocnemius muscle), and fast-twitch glycolytic (type IIX, white section of the gastrocnemius muscle) total muscle-tissue homogenates. Additionally, the rats were assigned to three different types of a run: (1) t = 30 [min], V = 18 [m/min] (group designation: M30); (2) t = 30 [min], V = 28 [m/min] (group designation: F30); and (3) t = 120 [min], V = 28 [m/min] (group designation: M120). At the muscular level, higher VO_2_ max translates to higher oxidative capacity, thus a speed of 18 [m/min] corresponds to approximately 65–70% of the maximum oxygen uptake (VO_2_max) in rats, while the speed of 28 [m/min] corresponds to approximately 82% of VO_2_max [26]. 

## 2. Materials and Methods

### 2.1. Animal Experiments

The experiments were carried out on two-month-old male Wistar rats (body weight: 261 ± 5 g). The rats were housed in stable conditions, a temperature of 22 ± 1 °C and a reversed light/dark cycle (12/12 h) were maintained in the animal quarters. The animals were fed ad libitum with a commercial pellet diet for rodents, and had free access to tap water. The project was accepted by the Ethical Committee for Animal Experiments at the Medical University of Bialystok, Poland (permission no.: 72M/2017, date: 31 October 2017).

The animals were divided into the following groups (*n* = 10 per group):sedentary control (Ctrl),a group that underwent a moderately intense run on a treadmill set at +10° incline with the speed of 18 [m/min] for 30 [min] (M30),a group that underwent a fast run on a treadmill set at +10° incline with the speed of 28 [m/min] for 30 [min] (F30),a group that underwent a moderately intense run on a treadmill set at +10° incline with the speed of 18 [m/min] for 120 [min] (M120).

The above exercise intensities were chosen because they recruit different proportions of oxidative, oxidative-glycolytic, and glycolytic fibers [27]. All rats were familiarized with the exercise conditions for 15 min daily during five consecutive days of the week preceding the final experiment. Immediately following the exercise, the rats were anesthetized with pentobarbital administered intraperitoneally in a dose of 80 [mg/kg of body weight], and the blood was sampled from the abdominal aorta in order to obtain serum for further assessments. The samples were frozen at –80 °C until further analysis. Next, samples of the left (LV) and right (RV) heart ventricles, the sternal part of the diaphragm (D), the soleus (S), as well as the red (RG) and the white (WG) sections of the gastrocnemius muscle were taken. The leg muscles are composed mostly of oxidative, oxidative-glycolytic, and glycolytic fibers, respectively [27,28]. The muscle samples were cleaned from any visible fat and connective tissue, rinsed with saline, blotted dry, and flash-frozen in liquid nitrogen. All analyses were performed in the total muscle-tissue homogenates. 

### 2.2. Western Blot Analysis

A routine Western blotting procedure was used to examine protein expression of endothelial lipase (EL), as it has been described previously [29,30]. Briefly, muscle samples were homogenized in RIPA buffer containing protease and phosphatase inhibitors (Roche Diagnostics GmbH, Mannheim, Germany). The protein concentration was measured using the bicinchoninic acid method (BCA) with bovine serum albumin (BSA) as a standard. Subsequently, homogenates were reconstituted in Laemmli buffer. The stable amounts of protein (30 µg) were loaded on Criterion™ TGX Stain-Free Precast Gels (Bio-Rad, Hercules, CA, USA) and then transferred to a polyvinylidene difluoride (PVDF) membranes. Next, the membranes were incubated overnight at 4 °C with the EL (Abcam, Cambridge, UK) and GAPDH (Santa Cruz Biotechnology, Dallas, TX, USA) primary antibodies in a dilution of 1:500. Thereafter, PVDF membranes were incubated with the appropriate secondary antibody conjugated to horseradish peroxidase (Cell Signaling Technology, Danvers, MA, USA). Thereafter, protein bands were detected using a ChemiDoc visualization system XRS (Bio-Rad, Hercules, CA, USA) and quantified densitometrically with Image Laboratory Software Version 6.0.1 (Bio-Rad, Hercules, CA, USA). Endothelial lipase expression was normalized to GAPDH expression (reference protein). Lastly, the sedentary control was set to 100 and the running groups were expressed relative to the control.

### 2.3. RNA Isolation and Gene Expression Analysis

Total RNA was purified using the NucleoSpin RNA Plus Kit with RNase-free DNase I treatment (Ambion, Thermo Fisher Scientific, Waltham, MA, USA), according to the manufacturer’s protocol (Macherey Nagel GmbH and Co.KG, Duren, Germany). Spectrophotometric measurements (A260/A280) were made to assess the quantity of the extracted RNA. Synthesis of the cDNA was performed using the EvoScript universal cDNA master kit (Roche Molecular Systems, Boston, MA, USA) and amplified by quantitative real-time polymerase chain reaction (qRT-PCR) using the LightCycler 96 System Real-Time thermal cycler with FastStart essential DNA green master (Roche Molecular Systems). Primer sequences used in this study: EL—forward primer (5′-3′) ACCAGAGTGGTGGGACGTAG; reverse primer (5′-3′) GGACAGCCTCCTGTTGATGTCyclophilin A—forward primer (5′-3′) TGTCTCTTTTCGCCGCTTGCTG; reverse primer (5′-3′) CACCACCCTGGCACATGAATCC

The following reaction parameters were applied: 15 s denaturation at 94 °C, 30 s annealing at 61 °C for EL and 59 °C for Cyclophilin A and 30 s extension at 72 °C for 45 (for Cyclophilin A) or 55 (for EL) cycles. Melting curve analysis was performed to verify PCR product specificity. Reactions were run in duplicates and the expression was analyzed using the relative quantification method modified by Pfaffl [31].

### 2.4. Determination of EL, Cytokines, Total and HDL Cholesterol Levels in Serum

Serum endothelial lipase (LIPG) and cytokines (IL-1β, IL-6, and IL-10) concentrations were determined using the Enzyme-Linked Immunosorbent Assay Kit from Cloud-Clone Corp. (Houston, TX, USA), according to the manufacturer’s instructions. The absorbance of EL and cytokines was measured spectrophotometrically at 450 nm by the use of a microplate reader (Synergy H1 Hybrid Reader, BioTek Instruments, Winooski, VT, USA) and its concentration was calculated from the obtained standard curves. The results are expressed in picograms per milliliter of serum. 

The kit for analyzing serum total and HDL cholesterol was obtained from Abcam (Cambridge, MA, USA). All measurements were performed according to the manufacturer’s instructions. 

### 2.5. Statistics

The collected data were analyzed using R ver. 3.6.3 (a programming language/environment for statistical computing: https://www.r-project.org/, accessed on 1 January 2021). First, prerequisite assumptions (normal distributions and variance homogeneity) for the applied methods were assessed using Shapiro-Wilk and Fligner-Killen tests. The data that passed the above were analyzed using analysis of variance (ANOVA) with the post-hoc pairwise Student’s *t*-test (with Benjamini-Hochberg multiplicity correction for the obtained *p*-values). Otherwise, the data were evaluated using the Kruskal-Wallis test with the post-hoc pairwise Wilcoxon test (with Benjamini-Hochberg multiplicity correction for the obtained *p*-values). Only the corrected *p*-values that were smaller than 0.05 were recognized as statistically significant. The data are presented as the mean and standard deviation or the median and inter-quartile range. The way of presentation is indicated in a figure/table legend. Pearson correlation coefficients (r) were calculated for HDL-C and EL as well as for pro- and anti-inflammatory cytokines and EL serum concentrations within a given group and irrespectively of group membership. The obtained *p*-values were corrected using Benjamini-Hochberg multiplicity correction.

## 3. Results

### 3.1. Endothelial Lipase’s Expression Pattern in the Vascular Beds of Rat Muscles

In an effort to characterize the expression pattern of EL between different muscles of Wistar rats, we analyzed the protein content in several distinct total muscle-tissue homogenates. Since the endothelium of white gastrocnemius muscle (WG) had the lowest EL protein expression, we decided to treat it as a reference point and set its expression level at 100 arbitrary units (AU). Other muscles are represented in relation to the WG (Figure 1). In the control animals, the highest expression of endothelial lipase was detected in the left ventricle (763 [AU]), which was followed by the diaphragm (662 [AU]) (Figure 1). The intermediate expression level of the protein was found in the whole homogenate of the red gastrocnemius muscle (341 [AU]), whereas relatively low levels were detected in the right ventricle (199 [AU]) and in the soleus muscle (183 [AU]) (Figure 1).

### 3.2. Treadmill Running Affects the Gene and Protein Expression of EL in Vascular Beds of Different Rat Muscles

#### 3.2.1. Left Ventricle (LV)

Endothelial lipase’s mRNA expression was markedly diminished in the rats that ran for 30 min regardless of the effort intensity (−86% and −88%, for M30 and F30 vs. Ctrl, *p* < 0.05, respectively, Figure 2A). On the other hand, the 30 min run with moderate speed caused a significant reduction in the cardiovascular components of the left ventricle’s EL protein level as compared to the control (−29%, M30 vs. Ctrl, *p* < 0.05, Figure 2B). In comparison to the group M30, the rats that run with greater speed (F30) had an elevated endothelial lipase protein level (+26%, F30 vs. M30, *p* < 0.05, Figure 2B), whereas the longer duration of the run (2 h) was accompanied by a significantly increased mRNA content for EL (+3.5 fold, M120 vs. M30, *p* < 0.05, Figure 2A). 

#### 3.2.2. Right Ventricle (RV)

In comparison to the control, decreased mRNA expressions for EL were found in M30 and F30 (−74% and −60% for M30 and F30 vs. Ctrl, respectively, *p* < 0.05, Figure 2C), while the rats that ran for 2 h had a decreased EL protein content in the vascular bed of the right ventricle (−28%, M120 vs. Ctrl, *p* < 0.05, Figure 2D). Additionally, the increased duration of the run was characterized by an increased transcript level for endothelial lipase in comparison to the 30 min run with moderate speed (+2.3 fold, M120 vs M30, *p* < 0.05, Figure 2C).

#### 3.2.3. Diaphragm (D)

The expression of EL mRNA was significantly reduced in both F30 and M120 (−59% and −60%, for F30 and M120 vs Ctrl, respectively, *p* < 0.05, Figure 2E). A decreased amount of endothelial lipase protein was found in the M30 group in comparison to the control animals (−17%, M30 vs. Ctrl, *p* < 0.05, Figure 2F). On the other hand, increasing the duration of the run to 2 h caused an increase in the EL protein level in comparison to the sedentary animals (+19% M120 vs. Ctrl, *p* < 0.05, Figure 2F). Additionally, we noticed some changes between the runs, namely, increasing the speed or duration of the run decreased the EL gene expression by ≥64% in comparison to the rats that run for 30 min with moderate speed (F30 and M120 vs. M30, *p* < 0.05 Figure 2E). Furthermore, a single bout of exercise substantially elevated EL protein content in the F30 and M120 groups as compared to the M30 group (+32% and +44%, for F30 and M120 vs M30, respectively, *p* < 0.05, Figure 2F). 

#### 3.2.4. Soleus

There was a tendency towards an increased endothelial lipase transcript level in response to the 30 min bout of exercise regardless of its intensity. However, due to the spread in the data points, it did not reach statistical significance (+51% and +58%, for M30 and F30 vs Ctrl, respectively, *p* > 0,05, Figure 3A). Similarly, also the protein expression of EL did not differ between any of the groups that underwent a run and the control (Figure 3B).

#### 3.2.5. Red Gastrocnemius

In comparison to the control group, we found that 30 min run with both moderate and high intensity reduced the expression of mRNA for EL (−42% and −47%, for M30 and F30 vs. Ctrl, respectively, *p* < 0.05, Figure 3C). Additionally, its (EL) protein content in the endothelium of the red gastrocnemius was significantly elevated in all the groups after a run (+34%, +39% and +35%; for M30, F30 and M120 vs Ctrl; respectively; *p* < 0.05; Figure 3D). 

#### 3.2.6. White Gastrocnemius

The data obtained for the total homogenate of the white gastrocnemius muscle show significant reductions in the EL gene levels with regard to M30 and F30 in comparison to the control (−60% and −48%, for M30 and F30 vs Ctrl, respectively, *p* < 0.05, Figure 3E). Accordingly, all the rats that underwent a run had decreased EL protein contents in comparison to the control animals (−24%, −30%, and −33%; for M30, F30, and M120 vs Ctrl; respectively; *p* < 0.05; Figure 3F). Interestingly, no changes between the rats after a single bout of exercise were noticed (Figure 3F).

### 3.3. Serum Endothelial lipase (EL), Total- and HDL- Cholesterol and Cytokines (IL-1β, IL-6, and IL-10) Concentration

The serum concentration of EL after the 30 min run with either speed did not significantly differ from the control (Figure 4A). On the other hand, the 2h exercise with moderate intensity resulted in over 60% decrease in the blood serum EL concentration as compared to the control group (p < 0.05). Neither total- nor HDL-cholesterol (HDL-C) content varied significantly between the studied groups (*p* > 0.05, Figure 4B,C). We did, however, observed a positive correlation between serum HDL-cholesterol and EL concentration (r = 0.34, *p* > 0.05, Figure 4D, Table 1). Moreover, when the correlation was tested separately for all the studied groups, a strong positive correlation between the two was observed in the control group (r = 0.91, *p* < 0.05, Figure 4E, Table 1).

Moreover, we also analyzed the serum concentration of both pro- (IL-1β and IL-6) and anti-inflammatory (IL-10) cytokines. Although the levels of the investigated cytokines remained relatively stable among the groups, we still noticed a downward trend for IL-1β in the rats that were subjected to a single bout of physical activity (Figure 5A–C). Furthermore, we determined correlations between serum EL and the above-mentioned cytokines for each group separately and collectively for all the data (Figure 5D– F, Table 2, Table 3 and Table 4). After adjusting for multiple analyses performed none of the correlations turned out to be statistically significant. 

## 4. Discussion

### 4.1. Expression Profile of EL in Whole Muscle Tissue Homogenates

Endothelial lipase (EL) is an enzyme secreted by endothelium, that is capable of HDL phospholipids hydrolysis [5]. Its expression was previously measured in many tissues, most of all in the lungs, kidneys, liver, or placenta [4]. Other tissues are considered to have a rather low EL expression and/or were rather scarcely studied. EL is not expressed in myocytes per se, however, the density of the vascular beds within the skeletal muscle correlates with the level of EL expression in the tissue [25]. Skeletal muscles represent around 40% of the body mass of an average young man with ordinary body composition. The muscles contain different fiber types, namely slow-twitch oxidative (type I), fast-twitch oxidative–glycolytic (type IIA), and/or fast-twitch glycolytic (type IIX). Type I and type IIA fibers are characteristic for richly capillarized muscles. Dense vascular bed bestows on them a distinguishing reddish hue and for this reason, they are called “red” muscles). On the contrary, the muscles composed mainly of type IIX fibers have very poor vascularization and as a result, they have a more pale appearance, hence they are designated as “white” muscles [28,32]. In line with the above, Murakami et al. investigated the phenotype of two different muscle types: a) the soleus (SOL) and b) the extensor digitiorum longus (EDL) [33]. The muscles are commonly used as representative examples of oxidative and glycolytic muscle fibers, respectively. The soleus muscle had both a greater capillary density (476.24 [cap/mm^2^] vs. 383.06 [cap/mm^2^] for SOL vs. EDL, *p* < 0.05) and a greater capillary to fiber ratio (8.0 vs. 5.4 for SOL vs. EDL, *p* < 0.05) [33]. Research results show, that most muscle capillaries are closed at rest but open during exercise [34]. This allows for increased blood perfusion of the tissue, which shortens the diffusion distance between a capillary and a myocyte. Since skeletal muscles constitute around 40% of body mass, it is reasonable to assume that a vast portion of body capillaries is located in the tissue. Therefore it is expected to be an important site of EL localization in the body. Additionally, the myocardium is characterized by a high oxidative demand. This is evidenced by ~70% extraction rate of oxygen from the coronary arteries [35] and an immense capillary density reaching on average 2156 capillaries per mm^2^ in the myocardial tissue of a Wistar rat [36]. Surprisingly, literature analysis reveals that the number of scientific investigations regarding EL expression in the endothelium of muscles is relatively limited. Moreover, the authors tend to determine EL expression in muscle tissue as a whole, rather than in a particular muscle [5,7,8,25]. For instance, we found only one paper [25] that mentioned the specific name of a muscle, whereas other authors settle for the general term “muscle” or “skeletal muscle” [5,7,8]. Moreover, the research was rather focused on the transcript (mRNA) and not protein levels [1]. This seems to be of particular importance since the common view is that the two tend not to change in parallel and one should assess a molecule protein content as well [37]. As an example, Shimokawa et al. reported a high level of EL transcript in the gut, lungs, kidneys, and spleen in rats [15]. Additionally, the authors detected low levels of EL mRNA in the thoracic aorta, liver, and skeletal muscle. Unfortunately, the researchers did not provide information regarding the type of muscle used for RNA analysis [15]. Given the above, we believe the present study to be the first one with such a detailed comparison of the EL protein expression in different muscle types. For the purpose of this analysis, samples of striated muscles were obtained from the sedentary animals and their expression was determined by the Western blot technique. The highest levels of EL were observed in the left ventricle (764 [AU]), and diaphragm homogenates (663 [AU]). Interestingly, the endothelium of the right ventricle was characterized by a relatively small EL content (199 [AU]). Among the remaining examined skeletal muscle samples the highest protein expression for EL was found in the red gastrocnemius (342 [AU]), followed by the soleus (184 [AU]) and the white gastrocnemius muscle homogenate (100 [AU]). 

The results seem to be rather in line with the common belief that EL level is higher in tissues characterized by high O_2_ consumption and rich vascularization, after all, EL is produced by endothelium lining blood vessels. The heart and diaphragm seem to fit nicely in this pattern since they are both tissues with high oxidative capacity, and uninterrupted rhythmical contractile patterns [38,39]. The right ventricle, however, resembles rather skeletal muscle in this respect. This might be caused by different workloads of the two ventricles, workloads that may well translate into distinctive metabolic demands of the two muscles [39]. A paper by Zong and co-workers seems to address the problem [40]. The authors indicated that there is a significant difference between the two heart pumps, with the left one extracting ~75% of oxygen at rest in comparison to only ~50% extraction for the right ventricle [40]. This indicates lower metabolic requirements of the right ventricle and indirectly points to possibly smaller vascularization of the muscle. Unfortunately, we did not find research that would directly compare capillary density between the right and left ventricle of Wistar rat’s heart, still, a paper by Olianti et al. could potentially shed some light on the topic [41]. In the study, the authors compared heart capillary systems of control (WKY) and hypertensive (SHR) rats. Visual inspection of the data in Figure 2 indicates that the left ventricle has somewhat greater capillary density than its right counterpart, although a definitive statement would require proper quantitative analysis [41]. Still, this could potentially explain the observed smaller EL protein expression in the endothelium of the right ventricle found by us (768 [AU] vs. 199 [AU] for LV and RV, respectively). 

Another counterintuitive finding is the EL protein expression between the analyzed hind leg muscles. We analyzed three different muscle tissues: oxidative (the soleus), oxidative-glycolytic (the red gastrocnemius), and glycolytic (the white gastrocnemius) [42]. Based on the metabolic type of the muscle, one would expect to observe the following EL expression pattern SOL > RG > WG instead of the actual RG > SOL > WG (Figure 1). It is hard to explain this difference (greater EL content in the RG than in the SOL) since the literature data for the muscles capillary density in Wistar rats seem to be missing. However, a suitable study was conducted by Andersen and Kroese on human subjects [43]. Surprisingly, although the soleus got a greater ratio of slow-twitch fibers (64% vs. 50%, for the soleus and the gastrocnemius, respectively), its capillary density was smaller (288 [cap/mm^2^] vs. 365 [cap/mm^2^] for the soleus and the gastrocnemius, respectively) [43]. If the relationship holds true also for Wistar rats then it would be a sound explanation of the observed by us phenomenon. Nevertheless, this theoretical mismatch, i.e., greater oxidative capacity of the soleus in relation to its lower capillary number seems to be puzzling. Perhaps the smaller number of the capillaries in the soleus is compensated by their superior spatial arrangement. Individual muscle fiber in the soleus is accompanied on average by 2.23 capillaries in comparison to 1.53 capillaries for the gastrocnemius [43].

### 4.2. Changes in EL Expression Induced by a Single Bout of Exercise

In this study, we examined the content of EL at both mRNA and protein level in the endothelium of different muscles encompassing a broad metabolic spectrum, form eminently oxidative, i.e., the left ventricle, to clearly glycolytic, i.e., the white gastrocnemius. In the next step, we decided to investigate the effect of different forms of a single bout of exercise on the before-mentioned expression of EL. In our previous study, we demonstrated that even a single run can differently affect the expression pattern of quite a few enzymes involved in the heart lipid metabolism, i.e., adipose triglyceride lipase (ATGL), CGI-58, and G0S2 (activator and inhibitor of the enzyme, respectively) in the left and right ventricle [39]. The results prompted us to study the expression of EL not only in the vascular beds of the left but also of the right ventricle. We observed that short (30-min-long) bouts of exercise with either speed (18 [m/min] and 28 [m/min]) reduced the expression of EL mRNA in both ventricles. The expression partially returned to normal after 2h of moderate exercise. This pattern seems to be followed by the EL protein level as well, especially in the case of the left ventricular tissue. In the case of the protein, however, it is less apparent, since a change does not always reach the level of statistical significance. A mechanism underlying such fast changes in the mRNA/protein expression remains unclear. A possibility exists that during the first 30 min of an exercise degradation of the mRNA/protein in the heart was much faster than its production. The balance between the two processes was restored when the exercise was prolonged for up to two hours.

The diaphragm muscle is, like the myocardium, a muscle contracting continuously throughout life. It is composed mostly of red, highly oxidative fibers [44,45,46]. Both the muscles (the heart and the diaphragm) are characterized by an increased workload during exercise. Interestingly, the pattern of the EL expression in the endothelium of the diaphragm is different than in the ventricles. Moreover, the changes in the EL protein content differ from the one in the mRNA expression (reduced amount of EL mRNA in F30 and M120 vs. increased EL protein levels in F30 and M120). This indicates that the accretions in the enzyme protein were not a consequence of the de novo synthesis but rather a reduction in its breakdown/removal. No obvious reasons for the differences in the EL expression patterns between the diaphragm and the heart tissues are apparent at this point. 

Visual inspection of the data in Figure 3 indicated different dispositions of the EL expression for the analyzed skeletal muscle homogenates. Due to the scarcity of literature data, it is hard to explain the phenomenon. However, whatever the underlying mechanism is, it should have something to do with different tissue involvement in the exercise bout. In line with that notion, Armstrong and Laughlin investigated blood flows in different tissues of trained and untrained Sprague-Dawley rats in response to a treadmill run with the speed of 30 [m/min] [47]. Surprisingly, the soleus muscle of the untrained rats was characterized by a relatively small blood flow change (+39%) in response to the exercise (111 [mg*min^−1^*100 g^−1^] vs. 154 [mg*min^−1^*100 g^−1^] for blood flow at rest (R) and after 15 min of the run (E15), respectively). The above is in contrast to significantly higher delta (+1.75 fold) for blood flow in the white gastrocnemius (12 [mg*min^−1^*100 g^−1^] vs. 33 [mg*min^−1^*100 g^−1^] for R and E15, respectively). The greatest alteration in blood flow (+3.55 fold) was observed in the case of the red gastrocnemius (60 [mg*min^−1^*100 g^−1^] vs. 273 [mg*min^−1^*100 g^−1^], for R and E15, respectively). Based on the above, one could expect a greater response in the EL mRNA/protein content to be found in the gastrocnemius than in the soleus muscle tissue, which is the case in our study. 

The soleus muscle is engaged mostly in low to moderate-intensity exercise [48]. In our study we employed runs with two different speeds: 18 [m/min] and 28 [m/min]. The above corresponds to ~70% and ~80% VO_2_max respectively [32,49] and are considered to be vigorous or high-intensity exercise [50]. This might be the reason for which neither exercise bout significantly affected the EL mRNA/protein expression in the vascular bed of the muscle. On the contrary to the soleus, the mRNA expressions in the red and white sections of the gastrocnemius behaved like in the heart ventricles. Interestingly, the EL mRNA expression was also reduced in the endothelium of the white gastrocnemius after 30 min of moderate exercise when the muscle was rather only mildly involved in the contractile activity [48]. It might suggest that extra muscle signals were responsible for the reduction in the EL mRNA in the muscle. Interestingly, each exercise bout increased significantly the EL protein content in the red gastrocnemius and reduced it in the white gastrocnemius tissue. The changes in the EL protein content in the red gastrocnemius were in opposite direction to the changes in the mRNA EL expression. The red gastrocnemius muscle is composed predominantly of type IIA fibers which are richly capillarized (capillary density is 365 [cap/mm2] compared to 288 [cap/mm2] for slow-oxidative soleus muscle) [43]. At rest, most of the capillaries are closed but they open during exercise. Of course, this increases blood flow through the muscle and therefore may be a stimulus responsible for an increased EL protein expression in the tissue. However, it is hard to explain such a quick increase in the protein level. EL is produced in its inactive form and stored in the ER as takes place in the case of LPL [51]. Next, it undergoes sub-sequential post-translational modifications (like glycosylation) [51]. This could technically explain a relatively fast (30 min) increase in the amount of EL protein (quick post-translational modification and maturation of the stored pre-EL). The reduction in the protein expression in the endothelium of the white muscle was accompanied by the reduction in the mRNA expression. This might suggest a reduction in the enzyme protein synthesis. The employed exercises probably did not affect the total content of the muscle protein [6]. Therefore, it should be expected that the metabolism of the endothelial protein was regulated by some tissue-specific factors. Obvious candidates for the before-mentioned factors are myokines, i.e., a set of skeletal muscle-derived biologically active cytokines [52,53]. Those cell signaling molecules are secreted into the bloodstream or can act in a paracrine fashion [54]. The myokines IL-6 and IL-1β were shown to participate in the elevation of the EL protein level in the serum [19,20] and endothelium [22]. We decided to directly test that notion. As depicted in Figure 5 neither IL-6 nor IL-1β level differed between the control and any of the groups that underwent a single bout of physical activity. This might be attributed to the too-short time frame of our study since Hirata et al. [22] indicated that it may take 3–8 h of exposure to IL-1β to observe significant changes in EL mRNA level. Another possibility for the observed by us decreased EL mRNA expression is the action of some other myokines, e.g., IL-10 or IL-1ra, two anti-inflammatory cytokines whose peaks follow IL-6 secretion [55]. This seems conceivable, given a rather well-established positive association between inflammation and endothelial lipase [20]. Additionally, the fact that exercise exerts a general anti-inflammatory effect had us expecting to observe a reduced level of EL mRNA [56]. We also tested that hypothesis. Surprisingly, the IL-10 level was relatively constant among all the analyzed groups (Figure 5). Nevertheless, the term myokines encompass around 3000 molecules [57], so it is likely that some other members of that class could affect the enzyme (EL) protein expression. Another support for local regulation of EL expression may come from the experiment by Hamilton et al. [58]. The authors showed that electrical stimulation of the common peroneal nerve, 4h daily by 28 days, increased the mRNA and protein expression, as well as the activity of lipoprotein lipase (LPL) only in the stimulated tibialis anterior muscle but not in the unstimulated contralateral one. A mechanism underlying that phenomenon remains unknown. However, LPL is produced by skeletal myocytes so that there is a direct link between the contractile activity and metabolism of the enzyme. EL is produced by endothelial cells themselves, so the direct link between contractile activity and EL metabolism is less obvious. However, contractile activity increases the production of myokines by the muscles [54]. Therefore one could postulate that this effect was mediated by the myokines. 

### 4.3. Serum Endothelial Lipase (EL) and HDL-Cholesterol Levels

Serum EL concentration is affected by several factors, such as pro-inflammatory cytokines [15,16,17,18,19,22], type 2 diabetes [11] metabolic syndrome, and obesity [59,60,61]. Literature data indicate that serum EL concentration is associated with serum HDL-C concentration [59,60,61]. Since endothelial lipase is a major determinant of serum HDL-C level, the study of its physiological role is of particular interest for the development of anti-atherogenic therapies (low level of HDL-C is a recognized risk factor for cardiovascular disease) [62,63]. An excellent example of such a therapeutic approach is physical exercise, a natural activity exerting a beneficial effect on blood serum lipid profile [62,64]. Physical activity not only leads to a mere increase in HDL-C concentration, but also to an increased HDL content in the blood (via influencing endothelial function). Nevertheless, to date, there are no data exploring the effect of acute exercise on serum EL concentration. Herein, we demonstrated that 2h run of moderate intensity significantly reduced serum EL concentration. It is hard to satisfactorily explain this finding due to the scarcity of the literature data. Still, a prolonged medium intensity exercise is expected to rely predominantly on lipids metabolism [65]. It is known that EL cleaves lipoproteins and provides lipids for the cells to consume [4]. Therefore, a prolonged exercise bout in M120 should translate into their greater uptake, partly due to the greater EL expression. Endothelial lipase in the blood is derived mostly from vascular endothelium [66,67]. Greater demand for EL in the tissue vascular bed could well lead to its smaller release/detachment into the bloodstream, hence the observed downregulation of EL in the serum of the rats from the group M120. In regards to the unchanged level of total cholesterol and HDL-C, we believe it was caused by an upsurge in lipids release to the bloodstream. In our previous experiment [39] we showed that a single bout of physical activity significantly increases the plasma level of FFA with the most pronounced change observed in the case of the 120 min long run. Such an upregulated HDL release into the blood during exercise is conceivable and it was previously reported in humans [68].

## 5. Conclusions

In summary, our study delivers a few novel findings.

We evaluated, presumably for the first time, endothelial lipase (EL) protein expression in the endothelium of a broad range of muscle tissue. We observed the following EL content: LV > D > RG > RV > SOL > WG. Two of the examined muscle tissues showed EL levels that were lower than expected based on their oxidative capacity, namely the right ventricle (RV) and the soleus (SOL). This was, however, supported by the literature data of the capillary density in a given tissue, as explained in detail in the discussion. Interestingly, we observed that even a single bout of exercise of relatively short duration (30 min) was capable of inducing changes in the mRNA and protein level of endothelial lipase, with a clearer pattern observed for the former, especially in the case of muscles more active during exercise (the red gastrocnemius muscle). Most of the investigated vascular beds of the muscles were characterized by a reduced mRNA/protein expression in both M30 and F30 groups (a 30 min run at different speeds). This was at least partially reversed by a longer run (M120). Although the exact mechanism for such phenomenon is not entirely apparent at this point, still some intrinsic local (paracrine) factors, like myokines, might be involved. This is in line with the fact that only the 120-min-long run was capable of evoking a decrease in blood serum EL protein concentration.

## Figures and Tables

**Figure 1 biomolecules-11-00906-f001:**
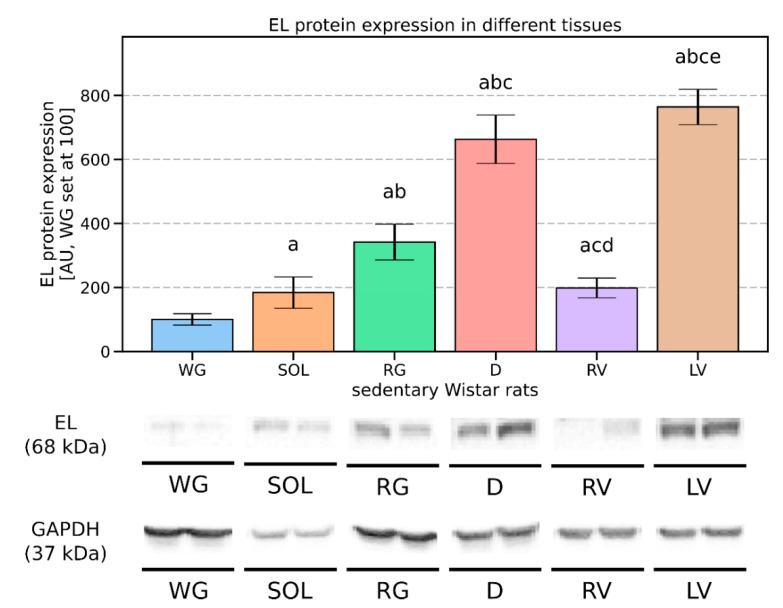
Endothelial lipase (EL) protein expression pattern in total muscle-tissue homogenates of sedentary control rats. Data are expressed as mean ± SD. For the sake of clarity, the EL expression level in the endothelium of white gastrocnemius muscle was set at 100 arbitrary units (AU), and the rest of the muscles are represented with respect to the WG. Representative Western blot images are shown. a—difference vs. white gastrocnemius muscle (WG), *p* < 0.05; b—difference vs. soleus (SOL), *p* < 0.05; c—difference vs. red gastrocnemius muscle (RG), *p* < 0.05; d—difference vs. diaphragm (D), *p* < 0.05; e—difference vs. right ventricle (RV), *p* < 0.05; *n* = 4 per group.

**Figure 2 biomolecules-11-00906-f002:**
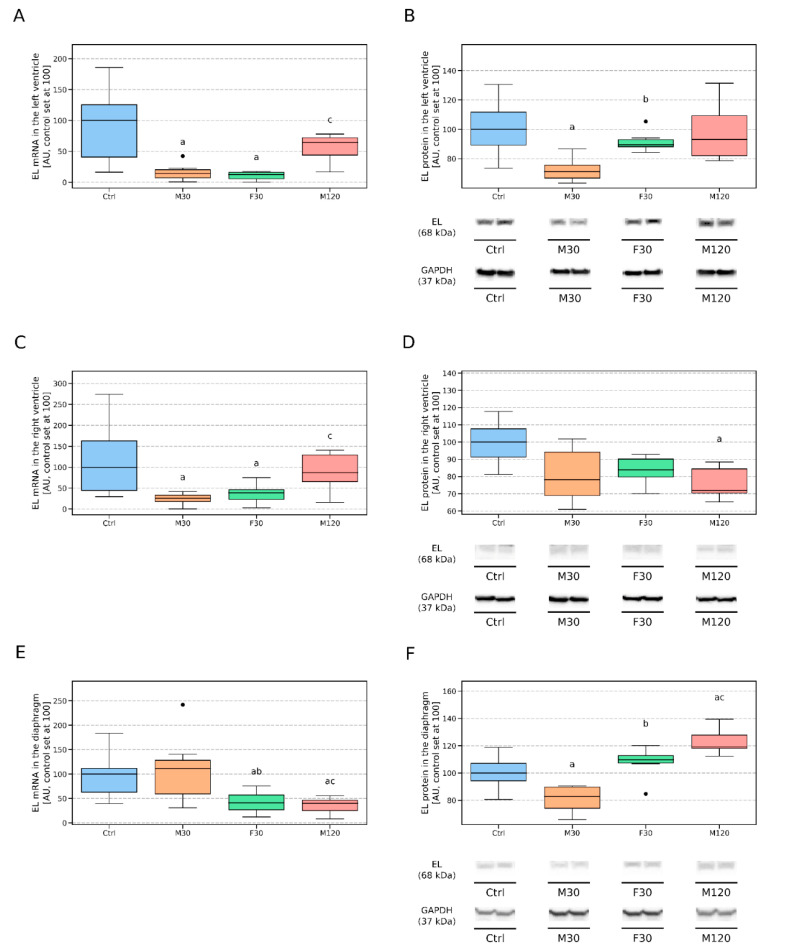
Endothelial lipase (EL) gene and protein expression in the vascular beds of the left ventricle (**A**,**B**), the right ventricle (**C**,**D**) and the diaphragm (**E**,**F**) in response to a treadmill run. The inner horizontal line of a box represents the median. Box boundaries: 25–75 percentile. Box whiskers: 1.5 interquartile range or max/min value in the group. Solid black dot—data point laying outside 1.5 * IQR (the interquartile range). Representative Western blots images are shown. a—difference vs. control (Ctrl), *p* < 0.05; b—difference F30 vs. M30, *p* < 0.05; c—difference M120 vs. M30, *p* < 0.05, *n* = 6 per group, RT-PCR assessments were performed in duplicates. Ctrl—control, M30—moderately intense run (speed: 18 [m/min], duration: 30 [min]), F30—fast run (speed: 28 [m/min], duration: 30 [min]), M120—moderately intense run (speed: 18 [m/min], duration: 120 [min]).

**Figure 3 biomolecules-11-00906-f003:**
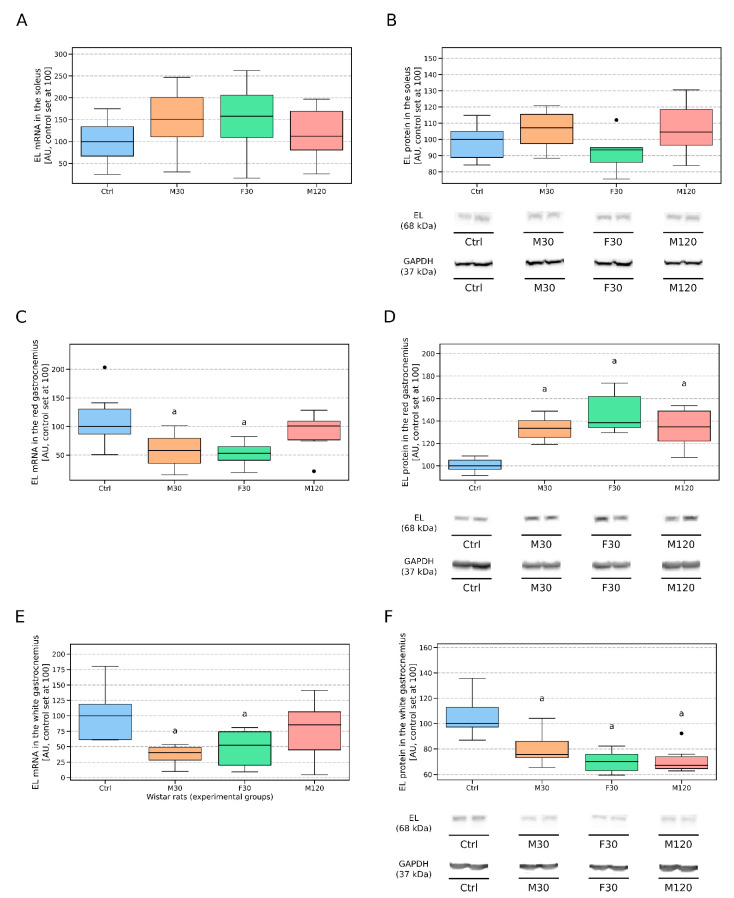
Endothelial lipase (EL) gene and protein expression in the vascular beds of the soleus (**A**,**B**), the red gastrocnemius muscle (**C**,**D**) and the white gastrocnemius muscle (**E**,**F**) in response to a treadmill run. The inner horizontal line of a box represents the median. Box boundaries: 25–75 percentile, Box whiskers: 1.5 interquartile range or max/min value in the group. Solid black dot—data point laying outside 1.5 * IQR (the interquartile range). Representative Western blots images are shown. a—difference vs. control (Ctrl), *p* < 0.05; b—difference F30 vs. M30, *p* < 0.05; c—difference M120 vs. M30, *p*  <  0.05, *n* = 6 per group, RT-PCR assessments were performed in duplicates. Ctrl—control, M30—moderately intense run (speed: 18 [m/min], duration: 30 [min]), F30—fast run (speed: 28 [m/min], duration: 30 [min]), M120—moderately intense run (speed: 18 [m/min], duration: 120 [min]).

**Figure 4 biomolecules-11-00906-f004:**
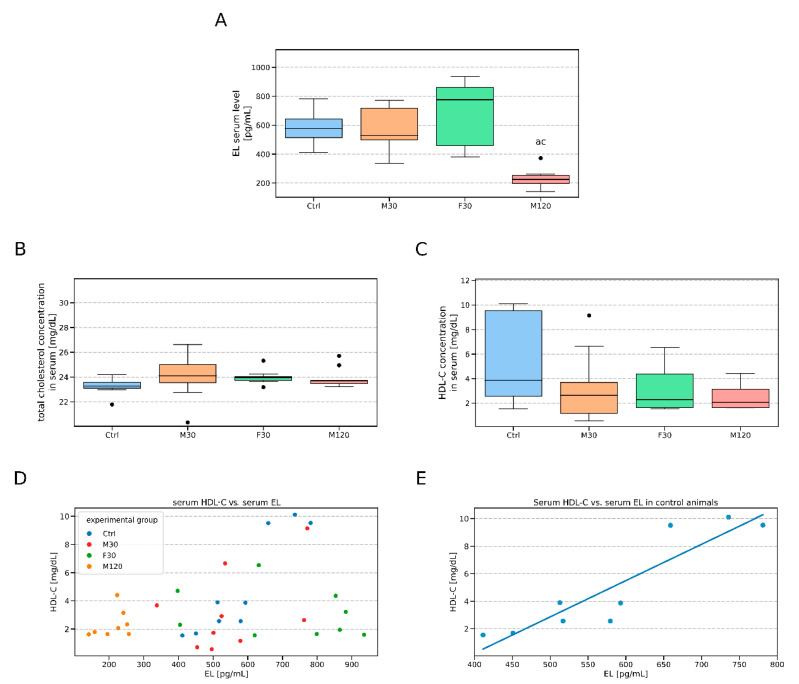
Serum endothelial lipase (EL) concentration and cholesterol profile in Wistar rats at rest and after a treadmill run. Serum endothelial lipase concentration (**A**), total cholesterol (**B**), HDL cholesterol (**C**), correlation between serum HDL cholesterol and EL concentration in all tested groups (**D**), correlation between serum HDL cholesterol and EL concentration in the control group (**E**). The inner horizontal line of a box represents the median. Box boundaries: 25–75 percentile, box whiskers: 1.5 interquartile range or max/min value in the group. Solid black dot—data point laying outside 1.5 * IQR (the interquartile range). The values are expressed in picograms per milliliter of serum. a—difference vs. control (Ctrl), *p* < 0.05; b—difference F30 vs. M30, *p* < 0.05; c—difference M120 vs. M30, *p*  <  0.05, *n* = 9–10 per group, ELISA assessments were performed in duplicates. Ctrl—control, M30—moderately intense run (speed: 18 [m/min], duration: 30 [min]), F30—fast run (speed: 28 [m/min], duration: 30 [min]), M120—moderately intense run (speed: 18 [m/min], duration: 120 [min]).

**Figure 5 biomolecules-11-00906-f005:**
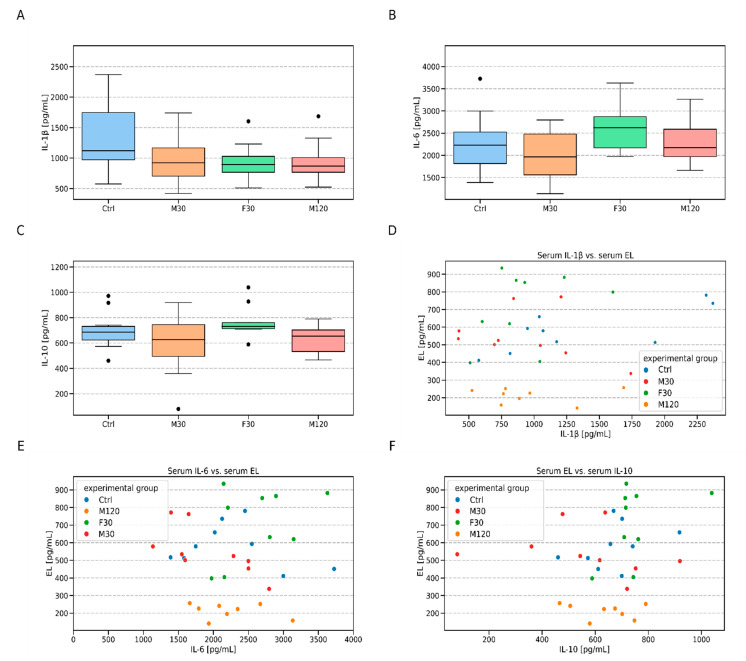
Serum pro-inflammatory IL-1β (**A**), IL-6 (**B**) and anti-inflammatory IL-10 (**C**) concentration. Correlation between serum IL-1β and EL concentration (**D**), correlation between serum IL-6 and EL concentration (**E**) and correlation between IL-10 and EL concentration (**F**) in all tested groups. The inner horizontal line of a box represents the median. Box boundaries: 25–75 percentile, box whiskers: 1.5 interquartile range or max/min value in the group. Solid black dot—data point laying outside 1.5 IQR (the interquartile range). The values are expressed in picograms per milliliter of serum. a—difference vs. control (Ctrl), *p* < 0.05; b—difference F30 vs. M30, *p* < 0.05; c—difference M120 vs. M30, *p* < 0.05, *n* = 10 per group. Ctrl—control, M30—moderately intense run (speed: 18 [m/min], duration: 30 [min]), F30—fast run (speed: 28 [m/min], duration: 30 [min]), M120—moderately intense run (speed: 18 [m/min], duration: 120 [min]).

**Table 1 biomolecules-11-00906-t001:** Correlation between serum HDL-C [mg/dL] and EL [pg/mL] concentrations.

Type of Analysis	r (Pearson Correlation Coefficient)	*p*-Value Unadjusted	*p*-Value Adjusted (BH)
Within Ctrl	0.910	0.001	0.005
Within M30	0.450	0.229	0.382
Within F30	−0.290	0.450	0.450
Within M120	0.350	0.402	0.450
All groups together	0.340	0.044	0.110

**Table 2 biomolecules-11-00906-t002:** Correlation between serum IL-1β [pg/mL] and EL [pg/mL] concentrations.

Type of Analysis	r (Pearson Correlation Coefficient)	*p*-Value Unadjusted	*p*-Value Adjusted (BH)
Within Ctrl	0.74	0.022	0.110
Within M30	−0.33	0.392	0.490
Within F30	0.35	0.352	0.490
Within M120	−0.02	0.955	0.955
All groups together	0.18	0.299	0.490

**Table 3 biomolecules-11-00906-t003:** Correlation between serum IL-6 [pg/mL] and EL [pg/mL] concentrations.

Type of Analysis	r (Pearson Correlation Coefficient)	*p*-Value Unadjusted	*p*-Value Adjusted (BH)
Within Ctrl	−0.29	0.455	0.583
Within M30	−0.70	0.037	0.185
Within F30	0.40	0.283	0.582
Within M120	−0.30	0.466	0.582
All groups together	0.06	0.737	0.737

**Table 4 biomolecules-11-00906-t004:** Correlation between serum IL-10 [pg/mL] and EL [pg/mL] concentrations.

Type of Analysis	r (Pearson Correlation Coefficient)	*p*-Value Unadjusted	*p*-Value Adjusted (BH)
Within Ctrl	0.37	0.326	0.543
Within M30	−0.27	0.486	0.586
Within F30	0.45	0.221	0.543
Within M120	−0.23	0.586	0.586
All groups together	0.19	0.283	0.543

## Data Availability

The data presented in this study are available on request.

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
