# Peer review of "Treadmill Running Changes Endothelial Lipase Expression: Insights from Gene and Protein Analysis in Various Striated Muscle Tissues and Serum"

_biomolecules, 2021, doi:10.3390/biom11060906_

Round 1

Reviewer 1 Report

The manuscript is significantly improved and deserves to be accepted in the present form.

Reviewer 2 Report

This manuscript is to study the effect of one-time exercise on the change of endothelial lipase (EL) in skeletal muscle. The data include the changes in mRNA and protein levels in sedentary and exercise animals and correlation of EL with lipid and cytokines, including HDL, IL-6 and IL-10. This manuscript provides some new and valuable information concerned with EL in skeletal muscle, however, there are some critical issues needed to be addressed.

  1. It is kind of strange that the tile is consisted of two not well-connected sentences.
  2. Although the authors claimed targeting EL in muscle in this study, they actually studied EL in cardiovascular components within muscle tissue. There is no information about EL in muscle fibers. What could be the direct functions of EL on muscle fibers?
  3. The authors used cyclophilin A as housekeeping gene, but previous study showed that exercise affected the expression of cyclophilin in skeletal muscle, how could this affect the results? Moreover, 45 cycles were programmed in RT-PCR experiments, what were the Ct numbers of EL expression?
  4. One of the important functions of EL is regulating HDL level. In Fig.4C, M120 group has significant lower level of EL, but no effect on HDL or total cholesterol, how to explain these results?
  5. The quality of Western blot results is poor. To better validate the data, whole blot pictures of Figures 2B, 2D, 2F, 3D and 3F should be provided to reviewers in next round review (it is not necessary to include these pictures in the manuscript).
  6. In Fig. 3C and D, it is unusual and questionable that mRNA level of EL decreased by 50%, while protein level increased by 40% after only 30 min exercise. Although the authors claimed that these effects could be resulted from changes in stabilities, how these happened and what kinds of mechanisms could contribute to these phenomena?

Reviewer 3 Report

The manuscript has been extensively improved, my previous concerns have been all amended.  

Round 2

Reviewer 2 Report

First of all, I would like to thank the authors for spending significant time and effort on answering my questions.

After reviewing the data, there are some critical concerns:

  1. In the first paper of endothelial lipase (EL) (PMID: 10192396), it is clearly stated that endothelial lipase is not expressed in skeletal muscle. By keeping this in mind, reorganization may be needed for this manuscript.
  2. In all the Western Blot data in the manuscript, the molecular weight of EL is 57kD (It is very possible that this is from the data sheet of EL antibody). However, in the original blot, it is 68kD, and it seems that the authors didn’t understand why there is a 10 kD difference.
  3. At least, I don’t think figure 2D is from the original blot of right ventricle. It is clear that in the original blot, although the signal of GAPDH is not good, but the bands are separated.

The authors did a lot of work and have improved the manuscript significantly. I suggest they double check the original data and do a resubmission.
